



# Co-variability drives the inverted-V sensitivity between liquid water path and droplet concentrations

Tom Goren[1,2], Goutam Chourdhury[1], Jan Kretzschmar[2], and Isabel McCoy[3,4]

[1]Department of Environment, Planning and Sustainability, Bar-Ilan University, Israel
[2]Institute for Meteorology, Leipzig University, Leipzig, Germany
[3]Cooperative Institute for Research in Environmental Sciences, Boulder, CO, USA
[4]NOAA Chemical Sciences Laboratory, Boulder, CO, USA

**Correspondence:** Tom Goren (tom.goren@biu.ac.il)

**Abstract.** Many studies using climatological data of liquid water path (LWP) and droplet concentration ($N_\mathrm{d}$) find an inverted-V relationship, where LWP increases and then decreases with $N_\mathrm{d}$. Our findings suggest that while these LWP responses to changes in $N_\mathrm{d}$ align with proposed causal mechanisms, such as entrainment evaporation feedback and precipitation suppression, the *inverted-V* is primarily driven by the co-variability between LWP and $N_\mathrm{d}$. This co-variability arises from meteorological con-
ditions and microphysical processes, each independently affecting LWP and $N_\mathrm{d}$ in opposite directions. We further demonstrate that the *inverted-V* relationship reflects the climatological evolution of Stratocumulus clouds (Sc). Therefore, background anthropogenic changes in $N_\mathrm{d}$ should, in principle, be manifested in changes across the entire Sc climatology along its evolution. Instantaneous LWP response to $N_\mathrm{d}$ derived from ship tracks, or other similar natural experiments, may therefore not accurately represent the climatological LWP response. This is because the local perturbations in $N_\mathrm{d}$ may not align with the plausible
natural co-variability between LWP and $N_\mathrm{d}$, which varies depending on the cloud state along the Sc evolution.

## 1 Introduction

Aerosol–cloud interactions are the greatest source of uncertainty in estimates of anthropogenic perturbations to Earth's energy budget (Forster et al., 2021; Boucher et al., 2013). Increases in atmospheric aerosols change cloud droplet number concentrations ($N_\mathrm{d}$), which in turn can change the cloud properties such as Liquid Water Path (LWP) and cloud cover. These changes,
known as "cloud adjustments" to aerosol perturbations (Albrecht, 1989), can lead to a significant cloud radiative forcing. However, the magnitude, and even the sign of this forcing are uncertain (Bellouin et al., 2020; Forster et al., 2021). Of great interest are the LWP cloud adjustments to $N_\mathrm{d}$, which have been shown to be positive, negative, or exhibit a weak variable response (Fons et al., 2023; Glassmeier et al., 2021; Gryspeerdt et al., 2019; Manshausen et al., 2023; Toll et al., 2019).

Several studies have shown consistent evidence of an *inverted-V* sensitivity of LWP to changes in $N_\mathrm{d}$ (Arola et al., 2022;
Glassmeier et al., 2021; Gryspeerdt et al., 2019; Mülmenstädt et al., 2024). The *inverted-V* emerges from joint histograms of $N_\mathrm{d}$ and LWP in the log-log space, in which each column is normalized so that it sums to 1 (See Figure 1a and Figure S1b). The *inverted-V* indicates two opposite sensitivity regimes of the response of LWP to $N_\mathrm{d}$, positive for precipitating clouds and negative for non-precipitating clouds.





The two sensitivity regimes align with the microphysical understanding of clouds' responses to aerosol perturbations. In precipitating clouds, an increase in aerosols leads to smaller droplets, which limit the efficiency of collision-coalescence and thus suppresses precipitation (Albrecht, 1989; Koren et al., 2014; Rosenfeld, 2000). As a result, more cloud water is retained in the cloud, and LWP increases. In non-precipitating clouds, smaller droplets associated with higher aerosol levels lead to a sedimentation-entrainment feedback, suppressing droplet sedimentation and enhancing radiative cooling at the cloud tops (Ackerman et al., 2004; Bretherton et al., 2007b). Additionally, they contribute to an evaporation-entrainment feedback, where smaller droplets experience faster evaporation timescales (Wang et al., 2003; Xue and Feingold, 2006). Both processes lead to a decrease in LWP by enhancing the entrainment of dry air from the free troposphere into the cloud layer.

These physical processes explaining the observed *inverted-V* sensitivity offer a convincing understanding of the causal mechanisms involved. However, alternative interpretations remain possible. An increasing number of recent studies suggest that the *inverted-V* is influenced by co-variations between LWP and $N_d$ or satellite retrieval biases (Arola et al., 2022; Fons et al., 2023; Glassmeier et al., 2021; Kokkola et al., 2024; Mülmenstädt et al., 2024). For instance, Fons et al. (2023) developed a methodology to remove confounding influences from large-scale meteorology and showed the importance of accounting for the co-variability between cloud depth and droplet size. This is in line with George and Wood (2010), who did a comprehensive study of the Stratocumulus clouds (Sc) in the South East Pacific Ocean (SEP). They showed that the influence of continental aerosols on the Sc is associated with synoptic conditions that favor a shallower Marine Boundary Layer (MBL), which caps the cloud top heights and thus limits the LWP, resulting in a negative correlation between LWP and $N_d$. Their study demonstrates how meteorology could complicate the interpretation of such correlations as being due to cloud response to aerosols. Similar insights come from Mülmenstädt et al. (2024), who used Global Climate Models (GCMs) and found that the MBL depth and $N_d$ in the Southeast Pacific (SEP) Ocean are anti correlated, which might explain the negative sensitivity regime of the *inverted-V*. Nevertheless, the response of LWP to $N_d$ across the entire range of the *inverted-V* is consistent with physical process understanding from theory and model simulations (Ackerman et al., 2004; Albrecht, 1989; Bretherton et al., 2007b; Koren et al., 2014; Rosenfeld, 2000). Here, we address and resolve this ambiguity.

## 2 Methods

We selected the major Sc regions of the Southeast Pacific, Northeast Pacific, Northeast Atlantic, Southeast Atlantic and Eastern Australia (see Table 1 for region limits). We focus on marine low level clouds in these Sc regions because they contribute significantly to the uncertainties in cloud-radiation interactions, especially when comparing models to observations (Christensen et al., 2022; Gryspeerdt et al., 2022; Neubauer et al., 2014).

We used instantaneous satellite observations of microphysical cloud properties from the Moderate Resolution Imaging Spectroradiometer (MODIS) instrument (Platnick et al., 2016), with a nadir resolution of 1 km by 1 km. The parameters used in this study include the corrected reflectance at $0.64\mu$m (R), cloud cover, cloud optical thickness ($\tau_c$), liquid water path (LWP) and cloud effective radius ($r_e$). $N_d$ was calculated from $r_e$ and $\tau_c$ following Grosvenor et al. (2018). We filter the MODIS scenes to include only single-layered liquid-phase clouds based on the MODIS cloud multi-layer flag and the MODIS cloud-phase





**Table 1.** Latitude and longitude boundaries of the analyzed oceanic regions. SEP - Southeast Pacific, NEP - Northeast Pacific, SEA - Southeast Atlantic, NA - North Atlantic, WAU - West Australia, SO - Southern Ocean.

| Region | Longitude | Latitude |
|---|---|---|
| SEP | $70°$E - $110°$E | $10°$S - $35°$S |
| SEP-Coastal | $75°$E - $85°$E | $10°$S - $35°$S |
| SEP-Remote | $140°$E - $150°$E | $15°$S - $30°$S |
| NEP | $120°$E - $140°$E | $25°$N - $40°$N |
| SEA | $0°$ - $25°$E | $10°$N - $20°$S |
| NA | $8°$E - $30°$E | $33°$N - $50°$N |
| AU | $80°$W - $115°$W | $25°$S - $40°$S |
| SO | $120°$W - $17°$W | $50°$S - $60°$S |

metric. We exclude pixels with sensor angles $< 55°$ and solar zenith angles $< 65°$, as those were shown to have retrieval-related uncertainties (Grosvenor et al., 2018). The filtered cloud properties were gridded into a uniform latitude and longitude grid of $1°$ by $1°$.

We define optically thin clouds as pixels that have a $\tau_c \leq 3$, as shown by (O et al., 2018a; McCoy et al., 2023). However, relying on valid $\tau_c$ retrievals alone might underestimate the optically thin cloud fraction because of failed $\tau_c$ retrievals (Cho et al., 2015). To overcome this, we follow the Choudhury and Goren (2024) approach in which we use a combination of MODIS-derived R and $\tau_c$ (for more details please refer to Choudhury and Goren (2024)).

ERA5 hourly data at a uniform latitude-longitude resolution of $0.25^{cire}$ by $0.25^{cire}$ (Hersbach et al., 2020) was used for
the sea surface temperature (SST) and the temperature at 800 hPa. We use these properties to calculate the marine cold-air outbreak parameter ($M$) (Fletcher et al., 2016; Kolstad et al., 2009), a measure of lower tropospheric stability defined as:

$$\text{M} = \theta_{\text{SST}} - \theta_{800} \tag{1}$$

where $\theta_{SST}$ and $\theta_{800}$ represent the potential temperatures at the surface and 800 hPa, respectively. Note that the surface refers to the sea surface and not the near-surface layer in the atmosphere. $M$ was found to be strongly correlated with the MBL depth
(McCoy et al., 2023; Naud et al., 2018, 2020), which is the motivation for using it in our study.

The Sc regime identifications are developed by applying the supervised neural network algorithm designed in Wood and Hartmann (2006) to MODIS LWP data. The algorithm uses the power density function and power spectrum of LWP to determine whether swath sub-scenes of 256km×256km fall into one of three categories: open cells, closed cells, with the remaining low clouds as cellular but disorganized. Sub-scene classifications are then re-gridded onto a $1°$ by $1°$ grid (Eastman et al.,
2024). The climatological occurrence of each type of regime is averaged in each grid point, from which the Red-Green-Blue (RGB) composites are derived. A gamma correction of 1.2 was applied to the red (closed cells) and green (open cells) to adjust the brightness of the image and improve its visual representation.



## 3  Results

Figure 1a shows the *inverted-V* pattern for the SEP Sc region, derived using a year of MODIS observations of LWP and $N_d$.
The mean longitude is shown in color and reveals a robust geographical dependence: low LWP and high $N_d$ near the coastal regions (longitude 75°-85°W, cool colors), and low $N_d$ and high LWP in the remote oceanic areas (longitude 85°-100°W, warm colors). This geographical co-variability can also be seen in the annual climatology of LWP and $N_d$ of the SEP, Figure 1b and Figure 1c, respectively. Similar geographical co-variability is manifested in the *inverted-V* across all other Sc regions (see Figure S2). Throughout the paper we selected the SEP as a representative region since this region hosts one of the most persistent Sc decks, on which many studies have focused in the past Wood (2012).

### 3.1  The negative sensitivity regime of the inverted-V

The negative sensitivity regime of LWP with $N_d$ is seen for $N_d$ values greater than 30 cm$^{-3}$. Observations of the Sc regions indicate that the MBL tends to be shallower near coastal regions and to deepen westward, as sea surface temperature (SST) increases (Eastman et al., 2017; Sandu et al., 2010; Wood, 2012; Wyant et al., 1997). Retrieving MBL depth from observations or reanalysis is challenging and subject to uncertainties (Eastman et al., 2017). We therefore follow previous studies and use a proxy for MBL depth, $M$, defined as the difference between the potential temperatures at the 800 hPa level and at the sea surface (McCoy et al., 2023; Naud et al., 2018, 2020) (See Methods). Figure 2a shows the mean $M$ for each bin in the LWP-$N_d$ joint histogram, revealing a clear gradient from right to left: Low $M$ (shallow MBL) is located on the right hand side, while high $M$ (deep MBL) on the left hand side. The MBL depth caps the cloud top height and thereby controls the vertical development of the clouds and their LWP, which is the vertically integrated cloud water. In accordance, lower (higher) LWP is associated with shallower (deeper) MBLs, as shown in Figure 2a. Recalling Figure 1a, shallow MBLs are prominent in the eastern SEP near the coastal regions whereas deep MBLs are predominant in the western SEP in the remote oceanic regions. The right to left gradient in $M$ therefore corresponds to an east to west gradient of MBL depth, which governs the climatological LWP. Similar results were recently shown by Mülmenstädt et al. (2024) in GCM simulations of the same region.

Comparing Figures 1a and 2a also reveals a pronounced east-west gradient in $N_d$. The higher $N_d$ levels in the eastern SEP are attributed to the proximity of clouds to continental aerosol sources, i.e., South America. In the SEP, as in other Sc regions, persistent south-easterly winds transport the high $N_d$ clouds from the coastal regions towards the remote oceanic areas downwind (George and Wood, 2010; Sandu et al., 2010; Wood, 2012). Along their advection, the clouds undergo a cleansing process through collision coalescence and droplet scavenging, causing a decrease in $N_d$ (Christensen et al., 2020; George and Wood, 2010; Goren and Rosenfeld, 2015; Goren et al., 2019, 2022; Rosenfeld et al., 2006; Yamaguchi et al., 2015). The concurrent opposing changes in LWP and $N_d$ can therefore also be explained by mircophysical processes, which would naturally lead to a negative relationship between LWP and $N_d$ (Gryspeerdt et al., 2022). Following this, the negative relationship between LWP and $N_d$ should therefore emerge in any approach that samples clouds throughout their temporal development, including large eddy simulations (LES) (Glassmeier et al., 2021), where clouds deepen over time. Aerosols





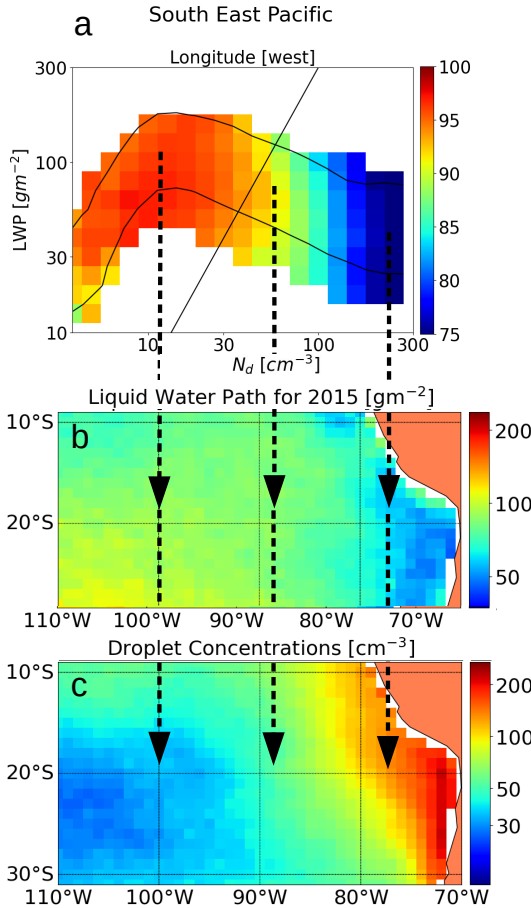

**Figure 1.** *inverted-V* and spatial variability of LWP and $N_d$. (a) Joint histogram of LWP and $N_d$ with mean longitude represented by color in each bin. The *inverted-V* emerges from the joint histograms of $N_d$ and LWP, where each column is normalized to sum to 1 (see also Figure S1). The black curves bound the bins with at least 10% of the column-normalized observations. The diagonal line represents an effective radius of 15 $\mu$m (assuming adiabatic clouds), serving as an approximate indicator of precipitation, with precipitating clouds located to the left of the line. Annual mean of (b) LWP and (c) $N_d$ over the SEP Ocean.

entraining from the free troposphere or originating from the ocean can influence the rate of decrease in $N_d$ (Wang et al., 2010; McCoy et al., 2024), and their climatology is assumed to be included in the climatological means presented here.

### 3.1.1 Temporal versus spatial cloud evolution

A measure for cloud droplet size is cloud effective radius ($r_e$). As clouds grow vertically, their cloud top $r_e$ increases accordingly (Freud and Rosenfeld, 2012; Gerber, 1996; Goren et al., 2019). Figure 2b shows the mean cloud top $r_e$ within each bin

of the joint histogram, where the $r_e$ can be seen to increase along the negative slope of the *inverted-V*. Also note that the $r_e$



**Figure 2.** Joint histogram of LWP and $N_{\mathrm{d}}$ showing the bin mean cloud properties for: (a) $M$, a proxy for MBL depth where a more negative $M$ indicates a shallower MBL, while a less negative $M$ indicates a deeper MBL. (b) $r_e$. (c) RGB composite of Sc cloud regime, where closed cells modulate the red, open cells modulate the green, and blue is modulated by other types of disorganized marine clouds. (d) Fraction of optically thin clouds (defined as cloud optical thickness smaller than 3). The black curves bound the bins with at least 10% of the column-normalized observations. The diagonal line represents an effective radius of 15 $\mu$m (assuming adiabatic clouds), serving as an approximate indicator of precipitation, with precipitating clouds located to the left of the line.

is approximately perpendicular to the $r_e$=15$\mu$m line. If LWP is increasing due to an increase in MBL depth (Figure 2a), the increase in $r_e$ can be attributed to the deepening of the clouds along their advection westward into the deeper MBL (Figure S4b shows $r_e$ versus $M$, where the consistent increase of $r_e$ with $M$ can be better seen).





Sandu et al. (2010) showed that the persistent winds in the Sc regions allow a time and space equivalence assumption. This
assumption was applied in Goren et al. (2022) to study the effect of aerosols on cloud cover, and can be similarly employed here.
Following these ideas, the climatological increase in LWP from east to west is in accordance with the simultaneous increase
in cloud top $r_e$, as if a single cloud is developing vertically over time. This time-space equivalence of cloud development was
applied by Gryspeerdt et al. (2021) to study the temporal evolution of ship tracks from instantaneous satellite observations and
by Rosenfeld and Lensky (1998) to examine the vertical profile of $r_e$ in convective cloud fields from instantaneous satellite
observations, which was found to be reliable based on LES simulation of Zhang et al. (2011). The above suggests that the
negative sensitivity regime of the *inverted-V* reflects also the clouds' temporal development across the Sc region, which is
comparable to the clouds' longitudinal (i.e., spatial) evolution.

### 3.1.2 Sc regimes across the inverted-V

The Sc evolution is also evident in the dominant Sc cloud regimes across the *inverted-V*. Figure 2c shows that closed cells are
most frequent where LWP is low and $N_d$ is high (red colors), transitioning into open cells where LWP becomes larger and
$N_d$ becomes lower (green colors). This aligns with the high occurrence of closed cells near coastal regions and the increasing
occurrence of open cells and other broken cloud regimes westward towards the remote oceans (Eastman et al., 2021, 2022;
McCoy et al., 2023; Muhlbauer et al., 2014). Furthermore, closed cells were shown in numerous studies to exist in high $N_d$
conditions and to break up when $N_d$ decrease sufficiently to initiate precipitation (Goren et al., 2019; Goren and Rosenfeld,
2014; Rosenfeld et al., 2006; Wang and Feingold, 2009), typically occurring at $r_e=15\mu$m (Freud and Rosenfeld, 2012; Gerber,
1996; Goren et al., 2019; Rosenfeld et al., 2012). This is consistent with the $r_e$ shown in Figure 2b and the lines marking the
$r_e=15\mu$m in Figure 2. It implies once again that the negative sensitivity regime of the *inverted-V* depicts the Sc evolution across
the region.

### 3.1.3 Local co-variability between air masses

The *inverted-V* is also found in spatially limited areas near the coastal regions, where the full climatology of the Sc evolution
across the entire region cannot be captured (Figure 3a and 3b). In these areas, the *inverted-V* emerges due to local temporal
variability in air masses. Although less frequent, clean air masses with deeper MBLs (low M, Figure 3a) can extend to the
coastal regions, resulting in anomalously deeper MBLs (and thus higher LWP), while shallow MBLs (and thus lower LWP)
dominate on most days. Since $M$ also characterizes air masses originating in the polar areas (Naud et al., 2018), the polar
maritime air might be relatively cleaner due to its origin and/or due to precipitation scavenging, and thus have lower $N_d$
despite being near the coastal region.

In contrast, the *inverted-V* pattern, particularly its negative regime, in spatially limited areas in the remote oceans is much
less pronounced and even non existing (see Southern Ocean example, Figure 3c and 3d). This is primarily due to the lack of
co-occurrence between high $N_d$ and low LWP (Figure S1c and Figure S1d). In distant Sc regions downwind the continents, the
occurrence of scenes with low LWP and high $N_d$ is significantly lower (Figure S5) and the negative sensitivity of LWP with





**Figure 3.** Joint histogram of LWP and $N_d$ for a small domain near the SEP coastal region ((a) and (b)), and for a small domain in the Southern Ocean ((c) and (d)). Refer to Table 1 for the boundaries of the regions. Panels (a) and (c) show the bin mean $M$, a proxy for MBL depth where a more negative $M$ indicates a shallower MBL while a less negative $M$ indicates a deeper MBL. Panels (b) and (d) show RGB composites of Sc cloud regime, where closed cells modulate the red, open cells modulate the green, and blue is modulated by other types of disorganized marine clouds.

$N_d$ diminishes accordingly. These findings elucidate the reason for the weak and positive LWP-$N_d$ sensitivities dominating the observations in remote oceanic regions as shown in Gryspeerdt et al. (2019).



## 3.2 The positive sensitivity regime of the inverted-V

The positive sensitivity regime of LWP with $N_d$ is shown in Figure 2 to the left of the $r_e$=15$\mu$m line (from which $r_e$ is larger).
Because $r_e$=15$\mu$m was shown to be associated with the onset of precipitation (Freud and Rosenfeld, 2012; Gerber, 1996; Goren et al., 2019; Rosenfeld et al., 2012), the clouds in the positive sensitivity regime are considered to be precipitating. A common explanation of the positive response of LWP with increasing $N_d$ is precipitation suppression. Precipitation suppression is the process where an increase in aerosols leads to an increase in $N_d$ accompanied by a decrease in droplet size, which in turn slows down collision-coalescence processes and inhibits precipitation formation (Albrecht, 1989; Koren et al., 2014), resulting in an
increase in LWP.

Figure 2a shows that the clouds in the positive sensitivity regime correspond to low $M$ (i.e., a deep MBL). Deep MBLs in the SEP are most common in the western part of the region, in accordance with our analysis shown in Figure 1 and previous studies (Eastman et al., 2017). However, despite existing in a deep MBL, it can be seen in Figure 2d that the optically thin cloud fraction, defined as clouds having a cloud optical thickness smaller than 3 (see Methods in Section 2), becomes larger
for less negative $M$ (deeper MBL) and lower $N_d$. This contradicts the intuition so far, since one would expect deeper clouds in deeper MBL. This contradiction is resolved when we consider the Sc regime evolution, as will be shown next.

Optically thin and ultra clean clouds have been found to exist at the top of deep MBLs (O et al., 2018a; Wood et al., 2018) as a result of lateral diverging outflows from active updrafts in precipitating convective elements. Due to coalescence scavenging in the precipitating updrafts, the cloud top outflows have low $N_d$ on the order of a few tens per cm$^3$ (Choudhury and Goren,
2024; O et al., 2018a; Wood et al., 2018). The latter studies have also shown that optically thin layers are most frequent in the eastern part of the SEP, in agreement with our observations (Figures 1a and 2d). Therefore, the higher occurrence of the optically thin layers in these regions is due to the deeper MBL which allows deeper clouds and thus precipitation, characterized with higher occurrence of open cells and other types of disorganized convection (Goren et al., 2023; O et al., 2018b; McCoy et al., 2023; Muhlbauer et al., 2014; Possner et al., 2020).

Figure 2c shows, accordingly, the dominance of open cells at the top of the *inverted-V* where active updrafts exist and form well defined open cell structures. Open cells, as well as other disorganized shallow convection regimes, dissipate eventually, leaving behind remnants of optically thin clouds with low $N_d$ (Choudhury and Goren, 2024; O et al., 2018a; Wood et al., 2018). This helps to explain the increase in optically thin clouds towards lower $N_d$ and lower LWP, suggesting that the positive LWP-$N_d$ sensitivity regime is in fact a continuation of the Sc regime evolution, decaying after the precipitating cores become
inactive. Such clouds are in their dissipation stage, not in their developing stage, and therefore the precipitation suppression mechanism cannot be applied to them. It should be noted that the *inverted-V* emerges also when restricting the scenes to cloud cover greater than 80% (see Figure S6), thereby reassuring that the sensitivities are not due to cloud microphysical retrieval biases in the broken cloud regimes (Cho et al., 2015; Painemal and Zuidema, 2011).





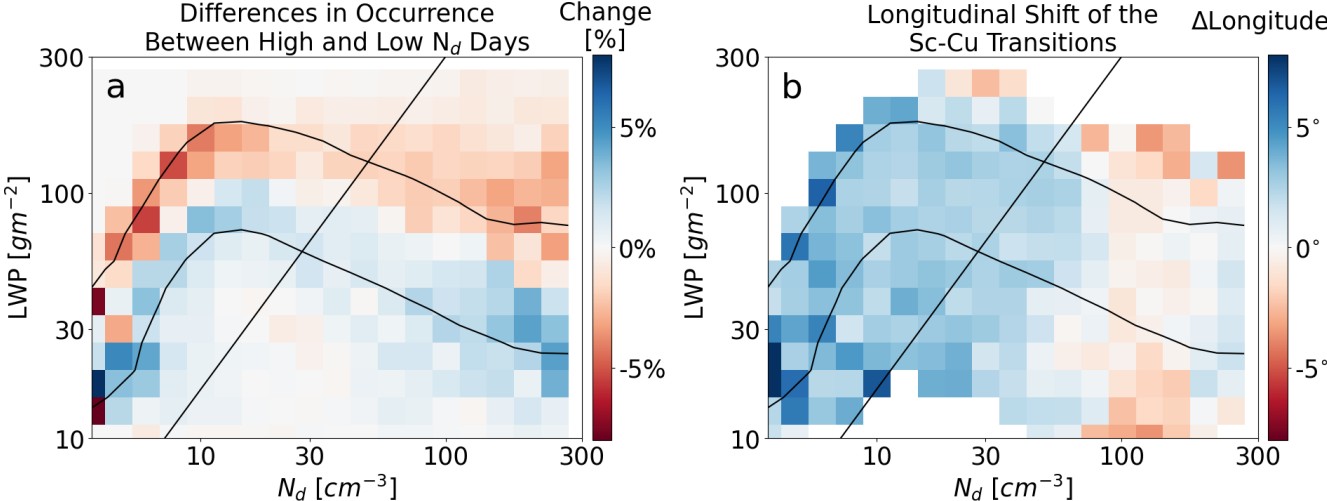

**Figure 4.** The effect of $N_d$ levels at the origin of the Sc region on the *inverted-V* climatology. (a) The difference in the occurrence between days with $N_d \geq 100$ gm$^{-2}$ and days with $N_d < 100$ gm$^{-2}$ near the coastal region (Longitude $<85°$W). The percentages represent the change in the normalized occurrence with respect to each column in the joint histogram, showing that the entire *inverted-V* shifts lower on days with high $N_d$ near the coastal region. (b) The mean change in longitude (in color) in each bin of the joint histogram between days with $N_d \geq 100$ gm$^{-2}$ and days with $N_d < 100$ gm$^{-2}$ near the coastal region (Longitude $<85°$W). The difference in both figures is calculated as high $N_d$ minus low $N_d$ days.

## 3.3 Synoptic co-variability between LWP and $N_d$

Sc typically form near the coasts and undergo a cleansing process as they are advected westward into the remote oceans. We
therefore divide the data into days with high and low $N_d$ at the coastal regions (defined as $N_d \geq 100$ cm$^{-3}$ and $N_d < 100$ cm$^{-3}$,
respectively, in Longitudes $<85°$W), assuming that near the coasts the initial $N_d$ of the clouds is determined, as shown by, e.g.,
Goren and Rosenfeld (2015). Given that synoptics vary slowly in the Sc regions (George et al., 2013; Goren and Rosenfeld,
2012, 2015; Sandu et al., 2010), we can assume that the clouds observed downwind of the coastal regions are a result of the
conditions that were observed upwind, i.e., near the coastal regions (Christensen et al., 2020; Goren et al., 2019; Gryspeerdt
et al., 2022).

Figure 4a shows the difference in the occurrence between the joint histograms of the high and low $N_d$ days. It can be seen
that high $N_d$ days favor low LWP across the entire *inverted-V* range (blue), and vice versa (red). This demonstrates the role of
the synoptic scale on the co-variability between LWP and $N_d$, as was also shown by George and Wood (2010) and Mülmenstädt
et al. (2024). George and Wood (2010) showed that for high $N_d$ days this co-variability in the SEP is a result of continental
air masses that are synoptically associated with shallower MBL, which limit the clouds vertical development and thus their
LWP. This emphasizes that caution must be taken when interpreting correlations as if they are causally-driven by aerosol-cloud
interactions.



Glassmeier et al. (2021) used an ensemble of LES simulations to show that the negative sensitivity regime of the *inverted-V* characterizes a steady state. The steady state is defined as a balance between evaporation at cloud top, which acts to lower LWP, and the build up of cloud water due to rediative cooling, which acts to increase LWP, with both being controlled by $N_{\mathrm{d}}$. The downward (upward) shift of the *inverted-V* when $N_{\mathrm{d}}$ is higher (lower) as shown in Figure 4a might therefore be related to the steady state hypothesis. In that sense, the steady state affects the *inverted-V* as a whole and does not necessarily shape it. Since Glassmeier et al. (2021) sampled clouds at different stages of development (i.e., across the cloud climatology), the co-variability between LWP and $N_{\mathrm{d}}$ is an inherent characteristic that arises from the temporal development of clouds and is therefore expected.

### 3.4 Climatological LWP adjustments

Figure 4b shows the *inverted-V* colored by the longitudinal displacement between days of high and low $N_{\mathrm{d}}$, as sampled in the coastal regions. It can be seen that the longitude at which $r_{\mathrm{e}}=15\mu$m (where transitions from closed to open cells typically occur) is shifted westward by up to 4 degrees. This means that on days of high $N_{\mathrm{d}}$ near the coastal regions, the Sc evolution is extended further west. This is expected since such days have lower LWP (Figure 4a), requiring the clouds to propagate further downwind into deeper MBL to gain LWP and to increase their $r_{\mathrm{e}}$ (Goren et al., 2019). This supports Gryspeerdt et al. (2022), who showed that the LWP response to $N_{\mathrm{d}}$ should be evaluated with respect to prior conditions.

Goren et al. (2019) used realistic Lagrangian LES to show that an increase in initial $N_{\mathrm{d}}$ in a given meteorological scenario delays precipitation formation and thus delays the closed cell breakup downwind, in agreement with our results. The delayed breakup was shown to allow the closed cells to deepen the MBL, which did not happen in the case of earlier precipitation and breakup. The role of $N_{\mathrm{d}}$ in deepening the MBL was shown in other studies (Bretherton et al., 2007a; Eastman et al., 2017) and was attributed to enhanced turbelence at cloud top due to stronger radiative cooling leading to greater entrainment (Bretherton et al., 2007a; Feingold et al., 1999). Following this, for the given climatological co-variability between LWP and $N_{\mathrm{d}}$, an increase in background $N_{\mathrm{d}}$ (e.g. due to anthropogenic activity) would lead to an increase in LWP because clouds deepens and gain higher LWP (Goren et al., 2019). Disentangling the change in the climatological LWP due to anthropgenic $N_{\mathrm{d}}$ is challenging from observations alone, as one needs to assume a counterfactual scenario with the exact co-variability between meteorology and aerosols (Goren et al., 2022). Because $N_{\mathrm{d}}$ influences the LWP via the Sc regime evolution, as shown here, adjustments in LWP and cloud fraction cannot be separated as they are closely interconnected.

### 4 Conclusions

The *inverted-V* sensitivity of LWP to $N_{\mathrm{d}}$, observed in numerous studies and often interpreted causally, is shown here to reflect the Sc regime evolution from overcast closed cells to open cells and subsequently to cumulus clouds and their ultimate dissipation. Studies that sample Sc at different stages of development, i.e., across their climatology, are therefore expected to populate the LWP-$N_{\mathrm{d}}$ joint histograms in an *inverted-V* shaped pattern (Arola et al., 2022; Dipu et al., 2022; Glassmeier et al., 2021; Gryspeerdt et al., 2019; Mülmenstädt et al., 2024; Possner et al., 2020).



The *inverted-V* is separated into two regimes, of negative and positive LWP-$N_d$ sensitivities. The negative sensitivity regime arises from the co-variability between $N_d$ and LWP. This co-variability is from two main components: (1) concurrent microphysical changes, where an increase in LWP is accompanied by a decrease in $N_d$, and (2) large-scale meteorology controlling the MBL depth and $N_d$, which vary in opposite directions simultaneously. The positive sensitivity regime reflects the dissipat-

ing stage of actively precipitating clouds, characterized by optically thin, ultra-clean cloud layers dominating the scenes (e.g., Wood et al. (2018)).

Our results therefore indicate that neither the negative nor the positive LWP-$N_d$ sensitivities that emerge in the *inverted-V* can be solely explained by causal effects of $N_d$ on LWP. Those causal effects are typically referred to as entrainment evaporation feedback for the negative sensitivities (Ackerman et al., 2004; Bretherton et al., 2007b) and precipitation suppression for

the positive sensitivities (Albrecht, 1989; Koren et al., 2014). Our results do not argue against the role of these physical mechanisms in the LWP response to $N_d$. Rather, our results demonstrate that even when a physical mechanism aligns well with observational correlations, care must be taken when interpreting these correlations as a means of inferring causality for aerosol-cloud interactions.

The *inverted-V* implies that the climatology of the LWP-$N_d$ co-variability has a plausible range in each geographical loca-

tion. A significant increase in $N_d$, such as from ship emissions (Goren and Rosenfeld, 2012; Manshausen et al., 2022, 2023; Toll et al., 2019) in areas where the background climatology of $N_d$ is lower, creates an instantaneous increase in $N_d$ that is detached from the plausible LWP-$N_d$ co-variability. Such $N_d$ perturbations reflect the causal, instantaneous LWP response and should not be generalized to the entire Sc climatological LWP sensitivity to $N_d$. This is in agreement with Glassmeier et al. (2021) who showed that such instantaneous $N_d$ perturbations overestimate LWP adjustments, as the climatology is in a steady

state whereas ship tracks are outside that steady state. We therefore suggest distinguishing between the causal instantaneous LWP response to $N_d$ and the climatological response to $N_d$. The former is relevant for studying LWP response associated with marine cloud brightening, in which aerosols are injected to increase the clouds' reflectivity (Feingold et al., 2024), while the latter pertains to changes in background anthropogenic $N_d$ levels, that is, the effective radiative forcing (Bellouin et al., 2020).

*Data availability.* All data sets used in this work are open source. The MODIS aqua cloud products are available from the Atmosphere

Archive and Distribution System (LAADS) Distributed Active Archive Center (DAAC): https://ladsweb.modaps.eosdis.nasa.gov/archive/allData/61/MOD06_L2/. ERA5 pressure level data were obtained from Copernicus Climate Change Service (C3S) Climate Data Store accessible at https://cds.climate.copernicus.eu/.

*Author contributions.* TG conceptualized the research idea, carried out the study, and wrote the initial manuscript. GC processed and co-located the datasets. IM contributed to the cloud regime classification. All authors contributed to the discussions and revisions of the

manuscript.



*Competing interests.* The contact author has declared that none of the authors has any competing interests.

*Acknowledgements.* TG acknowledges funding from the German Research Foundation (Deutsche Forschungsgemeinschaft, DFG; GZ QU 311/27-1) for the project "CDNC4ACI". GC and TG acknowledge startup funds from Bar-Ilan University. IM acknowledges support from the NOAA cooperative agreement NA22OAR4320151. The statements, findings, conclusions, and recommendations are those of the author(s) and do not necessarily reflect the views of NOAA or the U.S. Department of Commerce.




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
