# Peer review of "Co-variability drives the inverted-V sensitivity between liquid water path and droplet concentrations"

_EGUsphere, 2024_

## Author Comment (AC1)

We thank the reviewers for their careful reading of our manuscript. All of their suggestions have been thoroughly considered, and our responses are shown in red.

Reviewer #1:

This excellent paper addresses LWP-Nd covariability, a highly relevant scientific question within the scope of ACP. The Authors explain in detail how LWP-Nd covariability shapes the LWP-Nd inverted-V relationship. They demonstrate that the inverted-V relationship reflects the climatological evolution of Stratocumulus (Sc) clouds downwind from the coasts. They focus on low-level marine clouds in Sc regions and use MODIS satellite data, ERA5 reanalysis data, and Sc regime classification based on a neural network algorithm.

In my opinion, the conclusions are very well supported by the analysis, and the methods are fully appropriate. I want to thank the Authors for this important work highly relevant to improving the understanding of aerosol impacts on clouds and assessing aerosol forcing of Earth's climate. I strongly recommend the paper for publication in ACP and suggest a few minor points for the Authors to consider in revising the paper.

I suggest highlighting in the introduction that the inverted-V served as an important line of evidence in Ch7 of the Sixth Assessment Report by the Intergovernmental Panel on Climate Change for positive radiative forcing by decreased LWP (Forster et al., 2021) following Gryspeerdt et al. (2019).

This is now included in the second paragraph of the Introduction.

Moreover, I suggest more explicitly highlighting in the conclusions (and abstract?) that the inverted-V should no longer be used as a line of evidence for positive radiative forcing through LWP responses to aerosols as it is largely explained by the covariability.

As suggested, we included the above statement in the conclusions and in the abstract.

I suggest highlighting in the abstract the specific identified drivers of LWP-Nd covariability. Perhaps something similar to the first paragraph on page 12 under the conclusions?

Done.

What do you think about adding a table summarising the drivers of LWP-Nd covariability in Sc clouds (for two regimes, i.e. negative and positive LWP-Nd sensitivities)?

With the two drivers now explicitly mentioned in the abstract, we believe they will be more visible to skim readers as well.

I like Table 1. However, besides Table 1, a map with the region boxes would have been most helpful. Perhaps a map would work best in the supplement? A simple base map or a map with Nd and LWP climatologies?

A map with boxes indicating the regions listed in Table 2 is now provided as Figure S7.

p3l63 Perhaps the method by Choudhury and Goren (2024) could be summarised by a single sentence.

Done. The revised sentence is: "To overcome this, we follow the approach of Choudhury and Goren (2024), in which the observed relationship between R and successfully retrieved $\tau c$ is used to assign $\tau c$ to cloudy pixels with a failed $\tau c$ retrieval (for more details please refer to Choudhury and Goren (2024))."

p12l247 "Such Nd perturbations reflect the causal, instantaneous LWP response" I do not understand how a LWP response could be instantaneous. Do you mean with a short characteristic time scale, during which the steady state is not reached? Please note that "Instantaneous LWP response" is also mentioned in the abstract and p12l250.

Thank you for pointing this out. By 'instantaneous' we meant the LWP response to ship tracks, which differs from the aerosol effect on the Sc evolution. We have now removed the word 'instantaneous' to avoid confusion.

p1l9 the local perturbations in Nd may not align with the plausible natural co-variability between LWP and Nd; I am not sure what you mean by "align" here.

We meant that ship tracks are an anomaly in the LWP-Nd natural co-variability. We removed this part from the revised abstract because it is more of a hypothesis rather than a result. This is still being discussed in the discussion section. Instead, we have included the following sentence: "Therefore, background anthropogenic changes in Nd, occurring mainly near coastal regions where Sc decks initially form, should, in principle, be reflected in changes across the entire Sc climatology throughout its evolution."

**Technical corrections:**

I would suggest adding the depicted parameter with units next to the colour bar in all figures with colour bars.

The parameters and units are shown in the plots title. Including them alongside the colorbar would be redundant and would overcrowd the figure with text. We noted that units were missing in the supplementary material, and this has been corrected in the revised manuscript.

The titles of the main paper and supplement do not match:

"Co-variability drives the inverted-V sensitivity between liquid water path and droplet concentrations"

"Co-Variability, Not Causality, Drives Inverted-V Sensitivity Between Liquid Water Path and Droplet Concentrations"

Corrected. Thank you for noticing.

p3l64 0.25cire by 0.25cire -> circ symbol missing?

Corrected.

Which geographical region does Fig 4. represent?

The south east Pacific. This is mentioned in the revised caption:

" The effect of Nd levels at the origin of the Sc region on the inverted-V climatology in the SEP."

p3l63 we use a combination of MODIS-derived R and τc -> we use a combination of MODIS-derived R and τc to identify thin clouds

This sentence was revised according to a previous comment above referring to the same line. The revised sentence is as follows: "To overcome this, we follow the approach of Choudhury and Goren (2024), in which the observed relationship between R and successfully retrieved τc is used to assign τc to cloudy pixels with a failed τc retrieval (for more details please refer to Choudhury and Goren (2024))."

p4l85 Wood (2012) -> (Wood, 2012)

Corrected.

The caption of Fig4: Nd vs gm−2; please correct the units.

Corrected.

Fig S4 Could add the region (SEP) in the title so it would be formatted similarly to other figures.

Done.

Fig 1: What is depicted by thick black arrows? Meridians?

Yes. The following sentence was added to the caption: 'The vertical dashed arrows indicate the meridians."

p12l249 "Glassmeier et al. (2021) who showed that such instantaneous Nd perturbations overestimate LWP adjustments" Please rephrase to be more explicit. E.g. you could say "underestimate the decrease in LWP".

The sentence was removed from the revised manuscript.

Abstract p1l2 decreases with Nd -> decreases with increasing Nd

Corrected.

p1l2 LWP responses to changes -> LWP responses to increases

Corrected.

p1l19 to changes in Nd -> to increasing Nd

Corrected.

p1l22 "The inverted-V indicates two opposite sensitivity regimes of the response of LWP to Nd" It might be good to rephrase so no causality would be indicated, e.g. "two opposite regimes for changes in LWP corresponding to increase in Nd?" readers might think of a causal response when you say response?

Changed to: " It indicates two opposite sensitivity regimes in the relationship between LWP and Nd: positive for precipitating clouds and negative for non-precipitating clouds."

p2l44 "the response of LWP to Nd"; readers might think of a causal response when you say response?

The paragraph reviews prior studies providing evidence for co-variability between LWP and Nd. The line you refer to aims to convey that, even though the inverted-V pattern can also be explained by a causal response (i.e., precipitation suppression and entrainment-evaporation), the causal response fits well the inverted-v. This introduces the reader to the ambiguity surrounding what the inverted-V pattern actually represents. Following this, the text refers to a causal response of LWP to changes in Nd. We slightly revised the text accordingly: "Nevertheless, the response of LWP to increases in Nd is consistent with physical process understanding from theory and model simulations (Ackerman et al., 2004; Albrecht, 1989; Bretherton et al., 2007b; Koren et al., 2014; Rosenfeld, 2000). Here, we address this ambiguity."

p2l50 "especially when comparing models to observations" - perhaps it would be better to say something like: "and contribute significantly to the differences between models and observations".

Changed to: "We focus on marine low-level clouds in these Sc regions because they contribute significantly to the uncertainties in cloud-radiation interactions and to the differences between models and observations".

p6l118 consistent increase of re with M -> consistent increase of re with less negative M

Changed accordingly.

p9l154 "regime of LWP with Nd" perhaps this could be rephrased using a more formal wording?

Changed to: "The positive sensitivity of LWP to increases in Nd is shown in Figure 2, to the left of the re = 15μm line (where re is larger)."

Figure S4. "Correlations derived from the LWP-Nd joint histogram bins of the SEP." Is "correlations" the best wording here? Relationships?

"Correlations" changed to "Relationships".

p2l46 "Here, we address and resolve this ambiguity." I'm not sure if "resolve" is appropriate here, as the problem is likely not fully resolved. Although you identify important drivers of the co-variability, additional drivers may also play an important role…

Agreed. Changed to: "Here, we address this ambiguity."

Fig S3. Would you please add units for M.

Done.

Reviewer #2:

How to reliably assess the sensitivity of warm-cloud liquid water path (LWP) to anthropogenic and natural aerosol changes is a long-standing issue in cloud-aerosol-radiation-climate interactions. This paper analyzes an 'inverted-V' relationship between LWP and cloud droplet concentration Nd seen in MODIS retrievals over subtropical stratocumulus regions. The authors argue that this relationship is a byproduct of the downwind deepening and microphysical evolution of the cloud regime rather than a useful indicator of the climatological sensitivity of these clouds to an aerosol perturbation.

As the authors point out, past interpretations of satellite-derived LWP-Nd relationships have been controversial because they rely on inferring causation from clever analyses of correlations, complicated by potential satellite retrieval bias in some cloud regimes. To me, this study runs up against this same challenge. It is an appealing, thoughtful analysis that provides a plausible interpretation of the observed LWP-Nd relationship. It is certainly worthy of publication, but it is mostly a story of meteorological *caveat emptor* rather than showing a better way to use satellite observations to infer cloud adjustments to anthropogenic aerosol effects on Earth's radiation budget.

The study does not aim to provide a better way to use satellite observations to infer cloud adjustments to anthropogenic aerosols, and certainly more work needs to be done. The main purpose of the study is to present an alternative explanation for the LWP-Nd inverted-V sensitivity. The latter served as an important line of evidence in Chapter 7 of the Sixth Assessment Report by the Intergovernmental Panel on Climate Change for positive radiative forcing due to decreased LWP (Forster et al., 2021). This highlights the importance of our work, as the inverted-V should no longer be used as a line of evidence for positive radiative forcing through LWP responses to aerosols, since it is largely explained by covariability.

The satellite retrieval methodology may be strongly affecting the results:

- Putting aside possible biases associated with subpixel cloud variability, the 'filtered' cloud properties are averaged onto a 1°x1° grid (Line 58-59). Are cloud-free pixels being included in this average? I assume not, but this is important to explicitly mention, because if so, cumuliform and open-cell regimes would be expected to favor the left side of the V, because a population of uniform shallow clouds would produce grid-scale LWP and Nd that both scale with the cloud fraction, and therefore scale linearly with each other. This would also produce a low bias in LWP and Nd, if these are then interpreted as representative in-cloud values.

Cloud-free pixels are not included in the 1x1 degree averages. This means that clear pixels, which have no retrieved cloud properties, are considered NaN rather than zero. In other words, the gridded cloud properties represent in-cloud averages. The text has been revised to clarify this explicitly: "The filtered in-cloud properties were gridded into a uniform latitude and longitude grid of 1◦ by 1◦."

- The arguments in this paper might be more convincingly made using the 1 km pixel-scale retrievals that suffer much less from issues of averaging over a spatially heterogeneous cloud field.

Using 1 km pixels cannot provide information on the overall cloud field, which is particularly relevant for stratocumulus clouds that exhibit various cloud field morphologies. For example, the

cloud cover at the 1 km pixel level is binary (either 1 or 0), and thus cannot capture information about broader cloud field characteristics, such as distinguishing between open and closed cells. The same argument applies to LWP, which varies spatially on the scale of kilometers, even within fully overcast cloud scenes. Consequently, 1 km pixels cannot capture the spatial variability of LWP across different cloud field structures. Please note that the gridded 1x1 degree data is based on 1 km pixel-scale retrievals.

**Specific comments**

L46: 'Here, we address and resolve this ambiguity' - this study, like others before it, suggest that the observed LWP-Nd relationships in different boundary-layer cloud regimes are strongly tied to macroscale controls like boundary-layer depth. What ambiguity have the authors newly resolved?

The ambiguity lies between two potential explanations for the observed relationships in the inverted-V pattern: (1) the cloud response to entrainment (negative slope) and precipitation suppression (positive slope), and (2) the co-variability between LWP and Nd. Both explanations align well with the observed LWP sensitivity to Nd. In this study, we address this ambiguity, namely, the two competing explanations for the inverted-V, by focusing on the less explored explanation: the co-variability between LWP and Nd. Please note that we removed "'resolved" from this sentence, since more research is needed to fully understand this ambiguity.

We would like to emphasize that for the positive slope we introduce a new explanation that differs from precipitation suppression. We suggest that the dominance of thin clouds lowers the scene mean LWP as Nd decreases. This alternative explanation has not been proposed in previous studies.

L64: $^{\circ}$

Corrected.

L85: (Wood 2012)

Corrected.

L100-101: On Lines 37-41, the shallow PBL near the coast was also mentioned as a cause of the high Nd, referencing George and Wood (2010).

George and Wood found that shallow PBLs are associated with high Nd levels, but are not the cause of the high Nd, as stated in lines 38-40 of our manuscript: "They showed that the influence of continental aerosols on the Sc is associated with synoptic conditions that favor a shallower Marine Boundary Layer (MBL), which caps the cloud top heights and thus limits the LWP, resulting in a negative correlation between LWP and Nd."

L106: typo - should be 'microphysical'

Corrected.

L106: 'The concurrent opposing changes…should emerge in any approach that samples clouds through their temporal development': I don't get this argument. As we see in pockets of open cells, drizzle microphysics alone would tend to produce lower Nd and lower LWP downstream, if it weren't for downstream deepening of the boundary layer.

The deeper the clouds, the higher the LWP. As clouds become deeper, the cloud droplets size also increases. Larger droplets lead to more efficient collision-coalescence processes, which in turn result in lower Nd. Therefore, deeper clouds would have lower Nd.

In pockets of open cells (POCs), LWP is higher and Nd is lower, as shown, for example, in Figure 13 of Smalley et al. (2022) (https://doi.org/10.5194/acp-22-8197-2022). Eastman et al. (2022) (https://doi.org/10.1029/2022JD036795) suggest that POCs form under stronger winds, which enhance moisture fluxes from the ocean, leading to higher LWP, lower Nd, and precipitation. This is in line with our explanation in L109: as clouds develop, LWP tends to increase, accompanied by a decrease in Nd. Consequently, sampling clouds at different stages of their development would show a negative relationship between LWP and Nd.

The paragraph was revised for clarifications: "Comparing Figures 1a and 2a also reveals a pronounced east-west gradient in Nd. The higher Nd levels in the eastern SEP are attributed to the proximity of clouds to continental aerosol sources, i.e., South America. In the SEP, persistent south-easterly winds transport the high Nd clouds from the coastal regions towards the remote oceanic areas downwind, where the MBL is deeper, leading to an increase in LWP (George and Wood, 2010; Sandu et al., 2010; Wood, 2012). During advection, the clouds undergo a cleansing process through collision coalescence and droplet scavenging, leading to a decrease in Nd (Christensen et al., 2020; George and Wood, 2010; Goren and Rosenfeld, 2015; Goren et al., 2019, 2022; Rosenfeld et al., 2006; Yamaguchi et al., 2015). The concurrent opposing changes in LWP and Nd can therefore also be explained by microphysical processes, which would naturally lead to a negative relationship between LWP and Nd (Gryspeerdt et al., 2022). Following this, the negative relationship between LWP and Nd should emerge in any approach that samples clouds throughout their temporal development, including large eddy simulations (LES) (Glassmeier et al., 2021), where clouds deepen over time, thus enhancing the reduction in Nd via collision-coalescence.

L119-127: This is just the Lagrangian view of downstream cloud-topped boundary layer development, which dates back well before Sandu (e.g. Riehl et al. 1951 QJRMS). But note that air is constantly circulating vertically through the boundary layer during this downstream development, and being modified by surface heat and moisture fluxes as well as entrainment from above - this is very different than an individual cloud element that is vertically developing.

Even in non-advected cloud development, such as in developing continental cumulus fields during the diurnal deepening of the boundary layer, a negative LWP–Nd relationship is expected to emerge. This is because collision-coalescence processes increase with an increase in LWP, resulting in a decrease in Nd. In the referred paragraph (L119-127) we draw an analogy to the stratocumulus region, where the MBL deepens downwind, assuming time-space equivalence. This is supported by Figure S4, where the cloud top effective radius increases with longitude, corresponding to the deepening of the cloud (cloud top effective radius increases with the vertical extent of the clouds, e.g., https://doi.org/10.1029/2020JD033720).

L148: 'non existing' -> 'nonexistent'

Corrected.

L149: Add 'from' after 'downwind'

Corrected.

L150-151 and Fig. 3c-d: I get that the Southern Ocean has a lot of high-LWP stratocumulus clouds which form in a different high-latitude synoptic regime with smaller M and less persistent subsidence. But I am not sure what point you are trying to make here with 'These findings elucidate…'? Are you trying to say that it is noteworthy that Figs. 3b and 3d look different? Is there a reason to think that they would look the same, given the diverse ways that low clouds form and evolve?

Figures 3c-d are different from those in the stratocumulus regions, due to different synoptic regimes with larger M. In Gryspeerdt et al. (2019), Figure 2 shows a positive or no LWP sensitivity to Nd in the pristine oceans away from the stratocumulus regions. Our Figures 3c-d demonstrate that this is due to a different set of co-variabilities between LWP and Nd. It shows that the negative sensitivity observed in the stratocumulus regions is not necessarily a causal response of LWP to Nd, but rather reflects the different meteorological conditions.

We revised the text accordingly: "It shows that the negative sensitivity observed in the Sc regions is not necessarily a causal response of LWP to Nd, but rather reflects differences in meteorological conditions. Accordingly, these findings explain the weak and positive LWP–Nd sensitivities that dominate observations in remote oceanic regions, as shown by Gryspeerdt et al. (2019)."

L175-183: This argument is appealing, but it would seem to me to apply better at the 1 km pixel scale than at the 1° grid scale (~100 km, much larger than the ultraclean anvil region of precipitating shallow cumulus clusters), which blends the properties of cumulus clouds and ultraclean layers and gives LWPs that are probably representative of neither of these cloud types.

In our response to the second comment in this document we explain why the 1 km pixel is not suitable for the analysis. It is the blend of cloud properties, including cores and anvils, that is captured in the 1x1 degree analysis, allowing us to capture the different cloud morphologies. Zhou and Feingold (2023) (https://doi.org/10.1029/2023GL103417) showed that large cells typically have a horizontal size of ~64 km. Since cells size is scaled with MBL depth, these cells are likely to be associated with ultraclean layers, which occur in deep MBLs (e.g., Wood et al., 2018; https://doi.org/10.1175/JAS-D-17-0213.1). Therefore, 100x100 km scenes, certainly not less, is needed for our analysis.

Line 191: How long is a typical high-Nd or a low-Nd period in your data? In the SE Pacific, it takes several days for PBL air to advect from the coastal region to the stratocumulus edge.

We found 169 days in 2015 with high Nd (defined as Nd>100), with an average period of consecutive high Nd days of 2.4 days. George and Wood (2010) (https://doi.org/10.5194/acp-10-4047-2010) showed that pulses of continental aerosols last several days, controlled by changes in the synoptic flow in the SEP region. The advection of the clouds westward takes 2-4 days.

Therefore, the signal of near-coastal aerosol levels should be manifested downwind. This is explained further in the manuscript: "Given that synoptics vary slowly in the Sc regions (George et al., 2013; Goren and Rosenfeld, 2012, 2015; Sandu et al., 2010), we can assume that the clouds observed downwind of the coastal regions are a result of the conditions that were observed upwind, i.e., near the coastal regions (Christensen et al., 2020; Goren et al., 2019; Gryspeerdt et al., 2022)."

We acknowledge that a comprehensive analysis that directly links the starting conditions and the cloud evolution downwind is needed to more accurately capture the downwind effect on the Sc. We are currently working on such a study that includes Lagrangian trajectories.

L192-198: Nice sensitivity study!

Thank you!

Line 208: Clarify: 'the longitudinal displacement' of what? Also, in the argument that follows, note that there are surely synoptic differences (e.g. in subsidence or inversion strength) between high-Nd and low-Nd days that also help control the Sc edge in addition to the drizzle threshold.

The sentence was revised accordingly: "Figure 4b shows the change in longitude between days of high and low Nd, as sampled in the coastal regions, in every LWP-Nd pair in the joint histogram. It shows that the longitude at which re=15μm (where transitions from closed to open cells typically occur) shifts westward by up to 4 degrees."

The co-variability between synoptic conditions and aerosols (George and Wood, 2013) suggests that certain meteorological conditions are more common on high aerosol days, while different conditions prevail on low Nd days. This will be explored in a future study using Lagrangian analysis.

L221: Should be 'anthropogenic'

Corrected.

L251: I agree with your argument, but even the climatological Nd distribution is not easily interpretable just an response of the cloud regime to some external aerosol perturbation - instead Nd and the clouds co-evolve, depending on synoptic regime, to produce the observed LWP-Nd relationship, e.g. the simple model of Wood et al. 2012, JGR, doi:10.1029/2012JD018305. We can still get useful observational tests of cloud adjustments to anthropogenic aerosol in regions like the NW Pacific downwind from China, where there have been large changes in anthropogenic aerosol over the satellite record.

Our argument holds as long as the meteorology is the same. We added this to the revised manuscript: "The former is relevant for studying LWP adjustments associated with marine cloud brightening, where aerosols are injected to increase the clouds' reflectivity (Feingold et al., 2024), while the latter addresses changes in background anthropogenic Nd levels on the Sc evolution (assuming similar meteorological conditions), i.e., the effective radiative forcing due to aerosol-cloud interactions (Bellouin et al., 2020)."

Table 1: Degrees W and E are reversed in the table. Also, does the definition of the Southeast Atlantic region really go to 10 N? If so, call it the Tropical Atlantic.

Corrected. Thank you for noticing. The coordinates of the SEA were incorrect. They were corrected to 25N-0N and 20W-12E.

**Figures:**

(1) What are called 'joint histograms' are actually 'bin-averaged plots'. A joint histogram would instead show the relative frequency of each bin.

Thank you for this comment. We now use the term "2D bin-averaged plots" instead of "joint histogram" when referring to figures that show mean values.

(2) In Fig. 1, please include an actual histogram vs. Nd and LWP so the reader can appreciate how much of the cloud distribution lies in the different parts of the V. I think Fig. 1 could also be strengthened by including a map of low cloud fraction, from which I think the reader could infer that the region with Nd < 40 cm-3 has more broken, cumuliform cloud for which the MODIS retrievals might be less well suited.

The histogram is given in S1. We now explicitly refer the reader to S1: "see Figure 1a, which shows a 2D bin-averaged plot based on the joint histogram presented in Figure S1." Figure 1 now also includes cloud fraction contours.

(3) You might want to note that the color relationships in Figs. 2b and 2d are expected from the functional dependence of cloud optical depth and effective radius on LWP and Nd for a homogeneous warm cloud layer. Related to this, is there any upper bound on the MODIS-retrieved reff?

Indeed, the following sentence was added to the revised manuscript in Section 3.1.1: "It is worth noting that the relationship between re and LWP is also expected from the functional dependence of cloud optical depth and re on LWP and Nd for a homogeneous warm cloud layer (Wood, 2006)."

MODIS retrieved re has an upper bound of 30 mu. We are currently working on a related project in which we quantify the influence of this upper bound on LWP and Nd, and how this is propagated to the LWP–Nd sensitivity.

(4) In the caption of Fig. 3, if the Southern Ocean region is as given in Table 1, that doesn't seem like a 'small domain' since it spans over 100 degrees of longitude.

It should be Southern Ocean, not a 'small domain'. This has been corrected in the revised manuscript. Thank you for noticing. We also spotted a typo in the table, the longitude range should be between 120E and170E (not 120E and 17E). This has been corrected as well.

(5) Fig. 3 would be more self-contained if the lat/lon boundaries of the regions were included in the titles above the two rows, e.g. 'Eastern South Pacific (10-35 S, 75-85 W).

Added.